# Maize-bean intercropping mediates reduction in arthropod intraguild predation better than low-intensity farming—Stable isotope evidence

Nickson Erick Otieno[1]*, James Stephen Pryke[2], Jonathan Mukasi[3]

**1** Zoology Department, National Museums of Kenya, Nairobi, Kenya, **2** Department of Conservation Ecology and Entomology, Stellenbosch University, Stellenbosch, South Africa, **3** Kakamega Environmental Education Project, Shinyalu, Kenya

\* neotieno@yahoo.com

## Abstract

Crop-field structural management for boosting arthropod pest bio-control is increasingly recognized as an environmentally sustainable alternative to chemical pesticides. However, how natural pest regulation outcomes may be undermined by intraguild predation among pest natural enemies is seldom investigated in cereal crops-fields. Here we use $\delta^{13}C$ and $\delta^{15}N$ stable isotope analyses to assess intraguild predation amongst five arthropod taxa, comparing this to their consumption of three pest taxa, and test how such patterns relate to farming system (low-intensity/conventional) or cropping method (monoculture/intercropping) across 15 small non-transgenic maize fields. Overall, consumption of pests exceeded intraguild predation, especially Lepidoptera contribution to wasp, predatory beetle and predatory bug diets. However, every predatory taxon integrated diet carbon from all the others, particularly from predatory bugs. Predatory beetles showed the strongest connection to pest consumption while predatory bugs had the strongest connection to intraguild diet carbon. In general, wasps, predatory beetles and spiders exhibited more significant orientation towards pest consumption while predatory bugs were more of intraguild predators, but ants incorporated both food-sources nearly proportionately. Regarding influence of cropping options, pest consumption exceeded intraguild predation in intercropped compared to monoculture farms while for farming system, low-intensity rather than conventional systems promoted higher consumption of Lepidoptera pests. Low-intensity farming also enhanced intraguild consumption of predatory bugs. By contrast, while conventional farming boosted beetle and bug pest consumption, it was also associated with enhanced intraguild predation overall. Generally, therefore, field-level maize-field structural complexity through intercropping may be more impactful than chemical-free farming for mediating intraguild predation and hence boosting natural pest suppression. These findings, the first to quantitatively compare multiple-taxa pest consumption to intraguild predation in maize-fields, are important

**Data availability statement:** All relevant data are within the paper and its Supporting information files.

**Funding:** The author(s) received no specific funding for this work.

**Competing interests:** The authors have declared that no competing interests exist.

in informing appropriate agronomic management interventions within cereal-crop farming landscapes to maximize top-down pest biocontrol.

## Introduction

Cereal grain crops are damaged by a wide variety of arthropod herbivorous pests worldwide [1,2], but particularly so in low-income tropical countries where in addition to, and perhaps because of significantly smaller farming operational scales, farmers are unable to sustainably afford chemical pest management [3]. Such small farming scales serve to further enhance vulnerability to pest herbivory impacts and potential environmental degradation resulting from perennial use of agrochemicals [4]. To overcome such challenges, cereal-field habitat management measures involving enhancement of structural complexity and landscape heterogeneity to attract pest natural enemies, along with minimal agrochemical use, are increasingly recognized among tropical small-holder farmers as the more sustainable alternative for natural management of cereal arthropod pests [5–8]. Such in-field structural complexity enhancement strategies, which often also include crop diversification, intercropping, crop rotation and maintenance of uncultivated strips and margins, are now considered significant in facilitating pest natural enemy spillover from neighborhood habitats into crop-fields to enhance natural pest regulation [9–13].

However expected top-down pest suppression outcomes from such field management strategies are often undermined if natural enemy populations fall below critical thresholds or if they are dominated by specialist feeders. Beneficial results may also fail to be realized if herbivorous prey availability fluctuates due to local environmental perturbations including disruptive agronomic practices such as use of agrochemicals for pest control, weed suppression or soil management [4]. These may jeopardise availability of shared herbivorous prey resources thus intensifying natural enemy predator competition and conflicts [2,14,15], potentially diminishing predator-herbivore trophic interactions [14,16]. This may consequently occasion increased likelihood of intraguild predation either directly [17–19], or through consumption of early developmental stages of rivals [20,21]. Therefore field management practices that maximize predator-herbivore rather than predator intraguild trophic interactions are of significant importance in promoting overall efficiency of pest biocontrol in agricultural systems [13,22–24].

Intercropping and low intensity farming such as organic field management are two of the most widely applied strategies for promoting natural crop pest control [25–29]. Specifically, cereal intercropping enhances local-scale habitat complexity and overall arthropod diversity thus enhancing trophic interactions between herbivorous crop pests sand their natural predators [30–34]. On the other hand, low-intensity farming, which essentially involves limited or no use of agrochemicals is associated with overall soil health and conservation, pest natural-enemy preservation and a boost in agro-biodiversity due to enhanced habitat structural heterogeneity [28,35,36].

Stable isotope analysis (SIA) of feeding linkages between consumers and their food sources – is a robust, time-integrated yet less sample-intensive and less

invasive but more quantitative, technique compared to classical field observation methods for characterizing trophic connections [37–40]. Unlike many alternative molecular methods in which consumer food-source connections are deciphered from a network of species-level feeding connections without the time dimension [41,42], SIA facilitates direct trophic mapping of linkages over time including at community levels [38]. Finally, when coupled with Bayesian mixing modelling procedures, the technique also enables quantitative assessment of relative importance of various food source options to consumers' diets [39]. In SIA analysis, consumers are typically enriched in $\delta^{15}$N by ~ 3.4‰ and in $\delta^{13}$C by 1~1.0 ‰ relative to their food source through metabolic fractionation [37]. The $\delta^{13}$C and $\delta^{15}$N values are parts per thousand and represent fractions between the respective heavier and lighter isotopes ($^{13}$C/$^{12}$C, and $^{15}$N and $^{14}$N) determined from the relationship, $\delta nX = [(Rsample/Rstandard - 1)] \times 1000$; where $\delta nX$ is the parts per thousand difference (‰) between the $nX$ isotope in the sample and that in the standard; $Rsample$ is the ratio of the heavier to the lighter isotope of the element carbon or nitrogen, and $Rstandard$ is the ratio of the heavier to the lighter isotope in the standard [37].

This study's overall aim was to apply analyses of $\delta^{13}$C and $\delta^{15}$N stable isotopes to assess patterns of herbivorous pest consumption and intraguild predation by five arthropod natural enemy taxa, and to predict how these vary across farming systems and cropping methods. The specific objectives were to: 1) Estimate proportional contributions of three pest and five arthropod natural enemy taxa to diets of the predators; 2) Investigate any patterns of preferential intraguild trophic linkages; 3) Assess influence of maize farming system (low-intensity versus conventional) and cropping method (monoculture versus inter-cropping with legume crops) on pest consumption and intraguild predation patterns across the study farms [43].

We hypothesized that: 1) Intercropping and low-intensity farming increase pest consumption and reduce intraguild predation pressure; 2) Prey selectivity amongst predator taxa is not taxon-specific.

## Materials and methods

### Ethics statement and consent

As the study did not involve any handling of vertebrates or human subjects, therefore the ethical approval requirement was waived by the Ethics Committee of the National Museums of Kenya. Informed consent was obtained verbally from all farmers before commencement of the study

### Study area and study farm selection

The study was conducted in Kakamega County in western Kenya (00° 11' 09''- 00° 26' 08''N and 34° 44' 30''- 34° 51' 26''E), across 15 small-scale maize fields ranging from 0.6 to 9.0 ha in extent. The fields were selected within a mid-elevation landscape dominated by subsistence rural agriculture, with staple crops being maize and beans, but also potatoes and various varieties of vegetables [44,45]. The bi-modally distributed rainfall ranges from 1 200–2 000 mm annually. Out of the 15 selected study farms, three were monoculture low-intensity, five intercropped low-intensity, four were monoculture conventional, and the remaining three were intercropped conventional fields (S2 File). Inter-farm distances were maintained at 500–600 m to ensure regional similarity in general abiotic characteristics, while also ensuring sampling independence by minimizing effects of inter-farm dispersal of mobile arthropods. Samples were collected once at each of three maize crop stages (from germination to first weeding); mid-crop (from second weeding through flowering to corn-ear formation); and at mature-crop (from corn hardening to harvesting) in the short-rain season and in the long-rain season. Sampling during both rainy episodes was intended to minimize potential bias on the temporal scale that might be associated with variations in herbivorous prey resource dynamics [46].

### Arthropod sampling

Arthropods were collected using both standard sweep nets (100 sweeps along transects in each farm), taking sweeps at five-pace intervals down the center of each farm, sweeping on maize or bean leaves. These were supplemented with

pitfall traps, of 70 mm diameter and 120 mm high plastic cups filled to one-third volume with 25% sodium chloride solution for preservation and maintenance of isotopic integrity [46,47]. Four replicates were randomly placed along a diagonal line running across each maize field, and along these lines, the traps were spaced at distance intervals that depended on maize field size and they were collected after three days. Samples were identified to species and morphospecies. For spiders identification was made to Araneae Order, and then pooled into five predatory groups namely, ants (Hymenoptera,); spiders (Araneae), predatory beetles (Coleoptera); predatory bugs (Hemiptera); and wasps (Hymenoptera); and three herbivorous groups, butterflies and moths (Lepidoptera), phytophagous beetles and phytophagous bugs (S1 File). All arthropod samples were oven-dried whole, at 60°C to constant mass before being ground whole into fine powder. Subsequently, a 5 mg subsample was collected for each group for three replicates per farm which were then transferred into tinfoil capsules for isotope analyses.

## Sample preparation and isotopic analyses

The arthropod samples were analyzed at the Environmental Isotope Laboratory of the iThemaba Laboratory for the Accelerator Based Sciences (LABS) in Johannesburg, South Africa, to test for signatures of $\delta^{13}C$ and $\delta^{15}N$ isotopes. The analyses were accomplished on a *Flash HT Plus* elemental analyzer coupled to a *Delta V Advantage* isotope ratio mass spectrometer by a *ConFloIV* interface combusting at 1 020°C (*ThermoFisher, Bremen*, Germany). The $\delta^{13}C$ and $\delta^{15}N$ values were expressed as fractions of international reference standards Vienna Pee Dee Belemnite and air, respectively[37.] The isotopic food source base was established using the signatures of the three herbivore taxa, but also those of the five predators so as to facilitate assessment of intraguild predation, this food source base forming the foundational isospace biplot of $\delta^{13}C$ and $\delta^{15}N$ on which diet carbon was transferred to the predator taxa [38].

## Data analyses

Data for samples collected from the two cropping seasons were all combined and analyzed together to give an integrated perspective for the whole year of maize farming. All arthropod samples collected using the various methods at each farm, were also combined.

## Food source contributions to consumer diets

The stable isotope mixing model procedure, *MixSIAR,* was employed to determine (consumers' diet compositions from the range of food options (39) and subsequently calculate relative contribution of each food source to the consumer diet, incorporating consumer-specific fractionation discrimination values or trophic enrichment factors in the calculation [38–40,47]. The model applies the Markov chain Monte Carlo (MCMC) procedure within the *JAGS* package in R [48] to estimate probability density functions of trophic-related habitat and other explanatory variables [40]. The trophic enrichment factors (TEFs) that we applied were averages relevant to herbivorous arthropod consumers from a review by Spence and Rosenheim [49] and also from Caut et al. [50]. Because none of the predatory arthropod taxa analyzed here were obligate scavengers or detritivores, the contribution of these two feeding pathways were not considered significant enough to determine overall trophic connections [51–53]. Furthermore the study did not entail quantitative analysis of direct predation and the other feeding pathways separately, and therefore interpretation of isotopic signatures is taken to account for all the pathway options.

## Role of cropping method and farming system

To compare trends in food source contributions to consumer diets, two habitat factors, 1) cropping methods (monoculture versus inter-cropping) and 2) farming systems (low-intensity versus conventional) were considered. The models were run separately for predator-pest (predation rates) and predator-predator (intraguild predation) trophic linkages, by applying uninformative priors and designing *Jags* models while specifying for residual and process error structures [54]. This was

followed by running the *Jags* model starting with a test MCMC chain (1000 length and 500 burn-in iterations), and building up MCMC chain lengths (chain length and burn-ins iterations of 1 000, 5 000; 50 000, 25 000; 100 000 50 000, respectively) until convergences on true posterior distributions of food source proportion contribution to consumer diets were achieved, confirmed using the Gelman-Rubin diagnostics [40]. The final outputs comprised of estimated proportions of food source contributions to consumer diets based in response to the explanatory factors (farming system and cropping method) in form of medians, means and 95% credible intervals [40].

## Results

### General intraguild predation and prey preferences

Although every predatory taxa incorporated diet carbon from each of the three pest taxa, each of them also contributed at least some diet carbon each of their intraguild counterparts albeit in varying proportions (Fig 1A, B). Predatory bugs contributed the most to such intraguild predation both as consumer and as prey, targeted mainly by wasps (Figs 1A, B and 2) while ants were the second most intraguild prey target especially by spiders (Table 1).

Regarding potential for pest biocontrol, predatory bugs exhibited stronger roles in intraguild predation than in pest consumption (Figs 2 and 3A) while ants appeared to consume pests and intraguild prey at nearly equal proportions (Figs 2 and 3A). Predatory taxa that showed stronger roles in pest consumption than intraguild predation were predatory beetles, spiders and wasps, with predatory beetles showing the best overall potential as pest biocontrol agents (Figs 2 and 3A).

### General pest consumption patterns

Among the three pest taxa, Lepidoptera were the most significantly consumed by the predators (Table 2, S1 Fig), especially by predatory beetles and wasps while phytophagous beetles were mainly consumed by predatory beetles and phytophagous bugs by spiders (Tables 1 and 2, Fig 3B, D, E).

Predatory bugs, wasps and ants were the most targeted for intraguild predation (Tables 1 and 2, Fig 3B). Conversely, although spiders were the most common intraguild predators overall, consumption of predatory bugs by wasps was the most significant form of this phenomenon (S2 Fig), followed by spider consumption of ants (Table 1). Nevertheless, spiders consumed ants more intensively than ants consumed spiders Fig 3B. In addition, predatory bugs showed the only significant potential for cannibalism by deriving almost half of its diet carbon from its own kind, even though spiders and predatory beetles also incorporated nearly equal intraspecific carbon as that from their most valuable intraguild prey namely ants and spiders, respectively (Table 1, Fig 3B and S2 Fig).

### Role of cropping method and farming system

In general, cropping method was significantly influential in determining general consumption patterns across the farms, with higher levels of pest consumption observed in intercropped maize but higher levels of intraguild predation in monoculture systems (Table 3, Fig 4A). Conversely for farming system, pest consumption rate was lower than intraguild predation on low-intensity farms though the two forms of consumption were not remarkably different in conventionally managed systems (Table 3, Fig 4B).

With reference to specific taxa, in the case of pest consumption, Lepidoptera and phytophagous bugs were more intensively consumed in intercropped maize (Fig 5A) whereas phytophagous beetles were more heavily consumed in monoculture systems (Fig 5A). On the other hand, only Lepidoptera pests were more targeted in low-intensity while both phytophagous beetles and phytophagous were more depredated in conventional farm systems (Fig 5A).

For intraguild predation, apart from predatory bugs which were consumed more intensively within inter-cropped farms, intraguild predation targeted all the other four natural enemies more intensively in monoculture systems than in inter-cropped farms (Fig 5B). Similarly, predatory bugs were more intensively depredated on low-intensity farms while all the other four natural enemies were depredated more on conventional farming systems (Fig 5B).

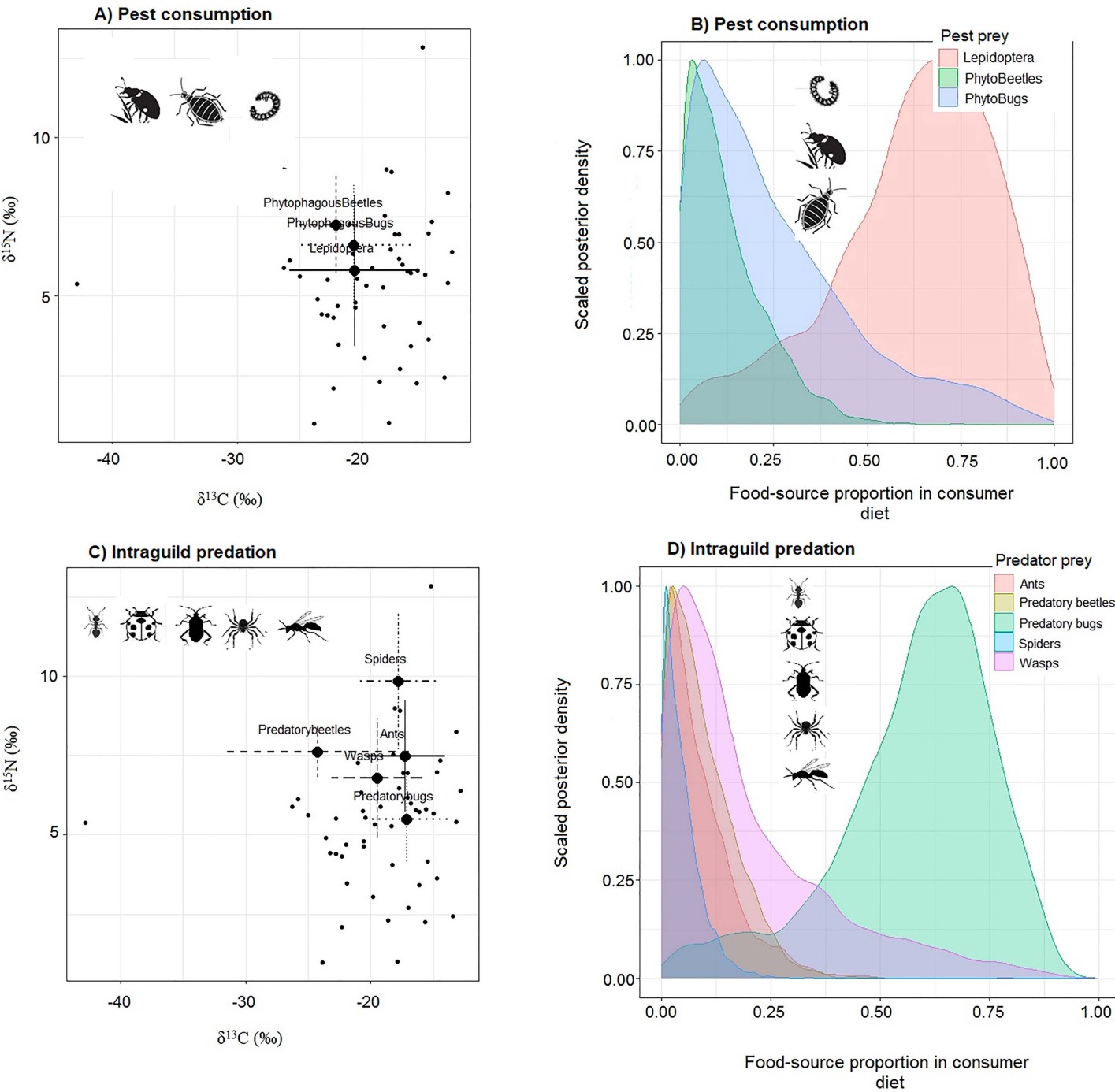

**Fig 1. MixSIAR modelling results showing general patterns of pest consumption and intraguild predation as depicted by Isospace plot of overall predator-prey association densities (A and C), and by Posterior density plots for the predators (B and D).** Plots indicate 95% credible interval overlaps in estimated proportions of food source contributions to consumer diets. *PhytoBettlles* = Phytophagoues beetles; *PhytoBugs* = Phytophagous bugs.

**Fig 2. Boxplots showing a summarized comparison of relative significance of each of the five predatory arthropod taxa as either pest consumers or as intraguild predators.** Plots are drawn from MixSIAR models of estimated proportional contributions food sources (pests and predators) to consumer diets. Box bars represent standard deviations (n = 3 for pest consumption and n = 5 for intraguild predation).

## Discussion

### General intraguild predation patterns

Every predatory taxon contributed at least some diet carbon to each of the others. This trend may not necessary indicate natural or sustainable trophic linkages in all these trophic interactions, but nonetheless implies that most of the predatory arthropods occasional shift their diets towards their more vulnerable conspecifics from time to time presumably when legitimate herbivorous prey are in short supply. A related study in orchards of Spain [55] also reported substantial quantities of carbon derived from intraguild prey than from herbivorous pets in diets of some spider species. Predatory arthropods may also derive intraguild diet carbon through consumption of carcasses of conspecific competitors [17,55], but also indirectly through consumption of herbivorous prey infested with parasitoid larvae [56–58], Dou et al. [58] showed that cotton aphids were the main medium through which parasitoid intraguild prey were consumed by ladybeetles in some parts of China.

Intraguild predation may also occur occasionally in the course of direct agonistic interactions such in territorial defence [17,59,60]. Additionally, for predators that supplement their regular feeding activities with detritivory, some small supplemental amounts of diet carbon may also be derived from soil humus and sediments bearing decomposed tissues of their dead counterparts [20,61], even if this might not make significant contribution to overall food intake.

Predatory bugs, however, appear to have contributed most substantially to diets of other predators, suggesting their relative vulnerability during intraguild interactions. This may have three possible explanations. First, apart from spiders

**Table 1. Relative consumption levels overall, of various pest and predatory arthropod taxa by their predators, indicating the two most significant consumers for each prey item. Values are median proportions of individual predators' tissues in the diets of the other predators, within 95% credible intervals (lower limit at 2.5% and upper limit at 97.5%).** *MSP = Most significant predator.*

| Trophic connection | Prey taxa | Lower credible limit | Median | Upper credible limit | Two most significant predators (MSPs) | Mean in MSPs diet |
|---|---|---|---|---|---|---|
| **Intraguild predation** | Ants | 0.003 | 0.054 | 252 | 1. Spiders | 0.166 |
| | | | | | 2. Ants | 0.076 |
| | Predatory beetles | 0.001 | 0.04 | 0.179 | 1. Predatory beetles | 0.091 |
| | | | | | 2. Spiders | 0.063 |
| | Predatory bugs | 0.016 | 0.32 | 0.649 | 1. Predatory bugs | 0.439 |
| | | | | | 2. Wasps | 0.366 |
| | Spiders | 0.001 | 0.029 | 0.129 | 1. Spiders | 0.176 |
| | | | | | 2. Ants | 0.056 |
| | Wasps | 0.002 | 0.069 | 0.399 | 1. Spiders | 0.098 |
| | | | | | 2. Predatory beetles | 0.072 |
| **Pest consumption** | Lepidoptera | 0.006 | 0.203 | 0.616 | 1. Wasps | 0.291 |
| | | | | | 2. Predatory beetles | 0.388 |
| | Phytophagous beetles | 0.002 | 0.051 | 0.249 | 1. Predatory beetles | 0.096 |
| | | | | | 2. Spiders | 0.081 |
| | Phytophagous bugs | 0.003 | 0.075 | 0.415 | 1. Spiders | 0.086 |
| | | | | | 2. Predatory beetles | 0.085 |

and ants, predatory bugs are among the most abundant and diverse pest natural enemies across agro-ecosystems [62,63] and thus more easily encountered by other predators, particularly given that many bug families exhibit mimicry of herbivorous arthropods [64,65]. Second, some families are also omnivorous and thus constitute legitimate prey to many predators [66] or as parasitoid hosts. Thirdly, as earlier explained, some predators may consumer eggs or the more vulnerable nymphal stages of predatory bugs, given the latter's wide distribution across the farm habitat [63,67]. For instance, Reeves [18] reported considerable consumption of predatory bug (*Anthocoris nemoralis*) nymphs by other predatory arthropods across pear orchards in the UK, including European earwigs (*Forficula auricularia*) despite presence of a wide variety of potential pest prey, the Pear psyllid (*Cacopsylla pyri*). Hambäck et al. [55] also observed that spiders consumed considerable quantities of larval and nymphal stages of bugs in orchards of Spain. Despite such intraguild tendencies, however, most arthropods predators generally obtain proportionately more of their nutrition from pest prey in many agroecosystems [20]. For instance, Lenka et al. [15] demonstrated that although most spiders that were active in Czech orchards during winter seasons consumed notable proportions of intraguild prey including some species of their own taxa, overall, they consumed proportionately more pest prey than their intraguild counterparts.

## Intraguild prey preferences

Other than predatory bugs being targeted mainly by wasps as intraguild prey, ants and wasps themselves were also significantly consumed by their conspecifics. As seen earlier, predatory bugs may be targeted by the others not only due to their wide field-wide distribution but also the characteristic omnivory among many families [62,63] but wasps may be consumed indirectly as larvae within their herbivorous hosts [58,67]. On the other hand, ants' considerable contribution to diets of other predators, particularly spiders may stem from ants' wide distribution, abundance, high mobility and diversity [30] which increase their likelihood as potential prey for spiders, especially the web-spinners which tend to be more abundant towards maturity-stages of maize growth [68,69]. Ants' symbiotic associations with certain herbivorous arthropods such as aphids, which they defend in exchange for certain sugars [70,71], may further render them vulnerable to

## A) Consumption overall

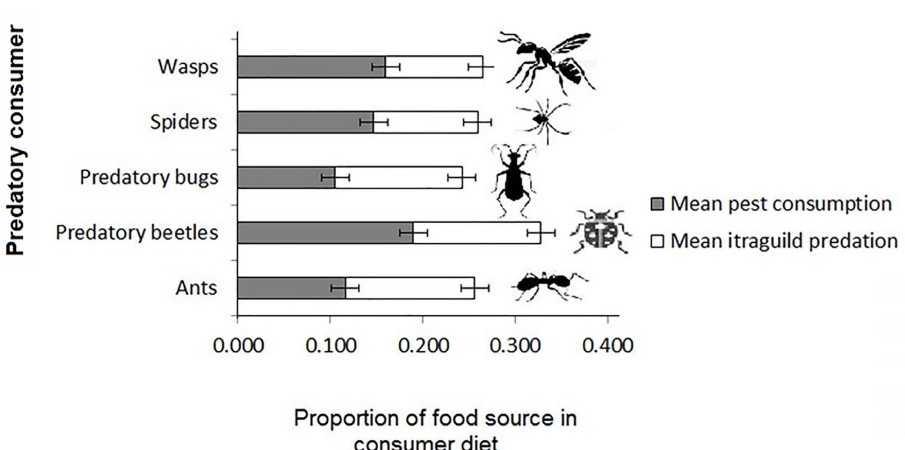

## B) Consumption pattern per prey type

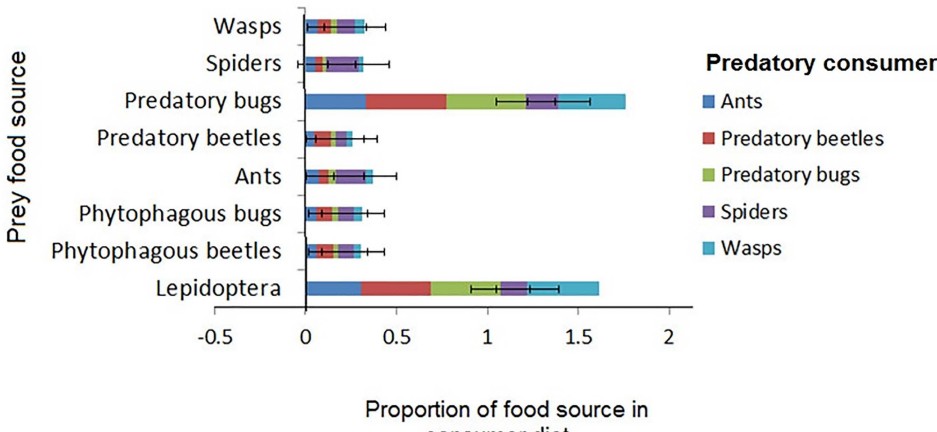

**Fig 3. Estimated dietary contributions of the three pest taxa and five arthropod natural enemies to diets of their respective predators: A) Overall and B) By each prey food source.** Box whiskers indicate standard errors, the top and bottom bars of boxes indicate 75th and 25th quartiles and the middle bars indicate mean values.

predation by some of their intra-guild competitors foraging for such aphid prey. Nonetheless, despite their trophic affinity to ants, spiders were also predicted to also incorporate considerable diet carbon from their own.

Furthermore, spiders emerged as the most liberal consumers overall, deriving nutrition not only from each of the pest taxa but also considerable diet carbon from the widest range of the predatory taxa, including their own. This is consistent with many previous studies across many habitats [20,21,72] and is attributable to spiders' high diversity, wide distribution and efficient hunting strategies. These include traits such as web-spinning among some families, for attracting a wide prey range [68,63,73]. Many spider families also exhibit comparatively localized foraging behaviour with relatively low dispersal often within clearly defined and actively defended territories so as to maximize foraging success [2,74,75]. By comparison,

Table 2. **Overall levels of pest consumption and intraguild predation regardless of predator identity. Values are median proportions of individual predators' tissues in the diets of the other predators, within 95% credible intervals.**

| Trophic connection | Prey item | Median | 95% Credible interval | |
| --- | --- | --- | --- | --- |
| | | | Lower credible limit (2.5%) | Upper credible limit (97.5%) |
| Pest consumption | Lepidoptera | 0.203 | 0.006 | 0.616 |
| | Phytophagous beetles | 0.051 | 0.002 | 0.249 |
| | Phytophagous bugs | 0.075 | 0.003 | 0..416 |
| Intraguild predations | Ants | 0.054 | 0.003 | 0.252 |
| | Predatory beetles | 0.041 | 0.001 | 0.249 |
| | Predatory bugs | 0.321 | 0.016 | 0.649 |
| | Spiders | 0.029 | 0.001 | 0.129 |
| | Wasps | 0.069 | 0.002 | 0.399 |

Table 3. **Summarized comparisons of proportional contributions of arthropod food sources to diets of their predatory consumers through the two trophic pathways of either pest consumption or intraguild predation, across the two treatment levels of farming system and cropping method.** *SD = Standard deviation.*

| Trophic connection | Agronomic practice | | 95% credible interval | | | | | |
| --- | --- | --- | --- | --- | --- | --- | --- | --- |
| | | | Median | | Lower limit | | Upper limit | |
| | | | Average | SD | Average | SD | Average | SD |
| Pest consumption | Farming system | Low intensity | 0.095 | 0.095 | 0.002 | 0.001 | 0.529 | 0.275 |
| | | Conventional | 0.118 | 0.086 | 0.006 | 0.005 | 0.437 | 0.174 |
| | Cropping method | Intercropping | 0.121 | 0.118 | 0.013 | 0.004 | 0.422 | 0.240 |
| | | Monoculture | 0.096 | 0.074 | 0.002 | 0.002 | 0.541 | 0.126 |
| Intraguild predation | Farming system | Low intensity | 0.088 | 0.133 | 0.002 | 0.003 | 0.388 | 0.287 |
| | | Conventional | 0.073 | 0.072 | 0.004 | 0.005 | 0.331 | 0.200 |
| | Cropping method | Intercropping | 0.097 | 0.119 | 0.003 | 0.002 | 0.309 | 0.234 |
| | | Monoculture | 0.073 | 0.072 | 0.001 | 0.002 | 0.464 | 0.179 |

ants derived the most substantial diet carbon from their intraguild counterparts, particularly predatory bugs which, as seen above, were also the most preferred intraguild prey by all predators. This may be attributable, more than anything else, to ants' often aggressive and highly gregarious foraging habits which is often instrumental in enabling ants to more easily subdue prey or predominate competitors [76]. Being frequent scavengers and detritivorous feeders, ants may also, like other scavenging arthropods, obtain intraguild diet carbon from carcases and eggs, and also, like some spider species [55,53], from soil humus with putrefied residues of other predators' body parts [20,61].

Interestingly, predatory bugs were notable as the only predators deriving significant proportions of diet carbon from their own kind, It is not clear why predatory bugs should obtain most of their nutrition through cannibalistic feeding behaviour, but this observation may suggest that intraspecific conflicts among predatory bugs such as those involving defence of resources, may be more frequent than those which occur amongst individuals of the other predatory taxa [16,61].

## Pest consumption

Lepidoptera were the most substantially consumed pest taxa overall by all predatory taxa, which indicates that it was the most preferred prey resource overall, mainly due to its widespread availability, detectability and occurrence at abundant occurrence in many crop-fields, especially in maize in addition to being among the easiest prey to handle for most predators [33,56,77–78,61]. By being the most significant predators of Lepidoptera, wasps and predatory beetles appeared to

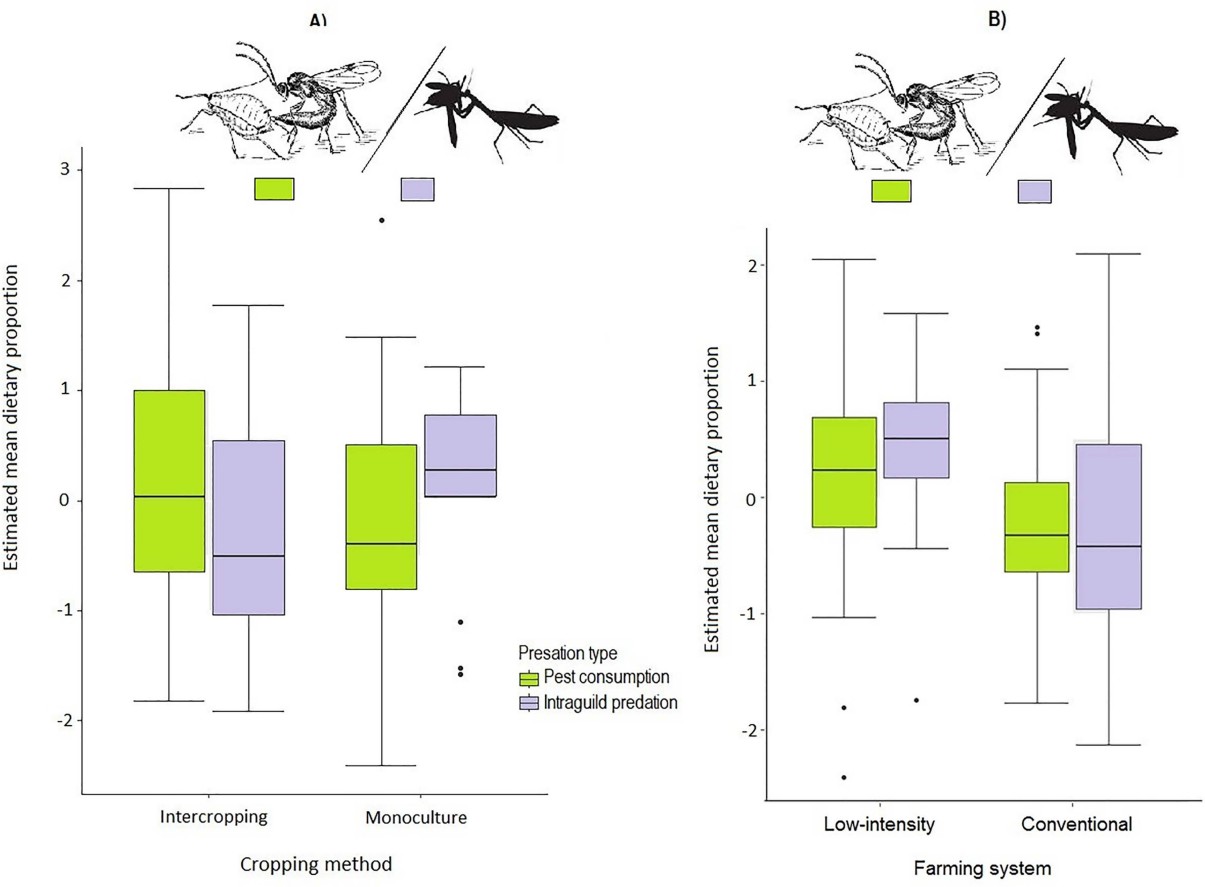

**Fig 4. Box-and-whisker plots showing estimated dietary contribution of the five arthropod natural enemies to diets of the respective intraguild consumers in response to: A) Cropping method and B) Farming system.** Box whiskers indicate standard errors, the top and bottom bars of boxes indicate 75th and 25th quartiles and the middle bars indicate mean values.

be potentially the most efficient biocontrol agents for this pest taxon across maize-fields. On the other hand, the significantly lower trophic connection between spiders and Lepidoptera as compared to that between spiders and either phytophagous beetles or phytophagous bugs, implies that the latter two pest taxa, due to their higher mobility and capacity for flight, may be more vulnerable than moth and butterfly larvae, to get entangled in webs spun by spiders [68].

### Influence of cropping method and farming system

Regarding the role of agronomic options, the first part of the second hypothesis postulating no effect of cropping method, was disproved, but not the second part relating to farming system. For instance, monoculture maize systems appeared consistent with higher levels of intraguild predation than pest consumption, contrary to the case for intercropping systems in which more pests were consumed than intraguild prey. This is attributable to the higher structural complexity of field intercrops, recognised for their role in facilitating enhanced pest consumption by their natural enemies [72]. This is in contrast to the more structurally simplified monocultures system associated with reduced diversity and abundance of arthropods in general [9,27]. Presumably owing to paucity of herbivorous prey diversity under such agronomic systems, for instance, some specialist predators and parasitoids may engage in more frequent agonistic intraguild conspecifics thus increasing chances of intraguild consumption [19]. Wasps, ants and predatory beetles appeared to be particularly

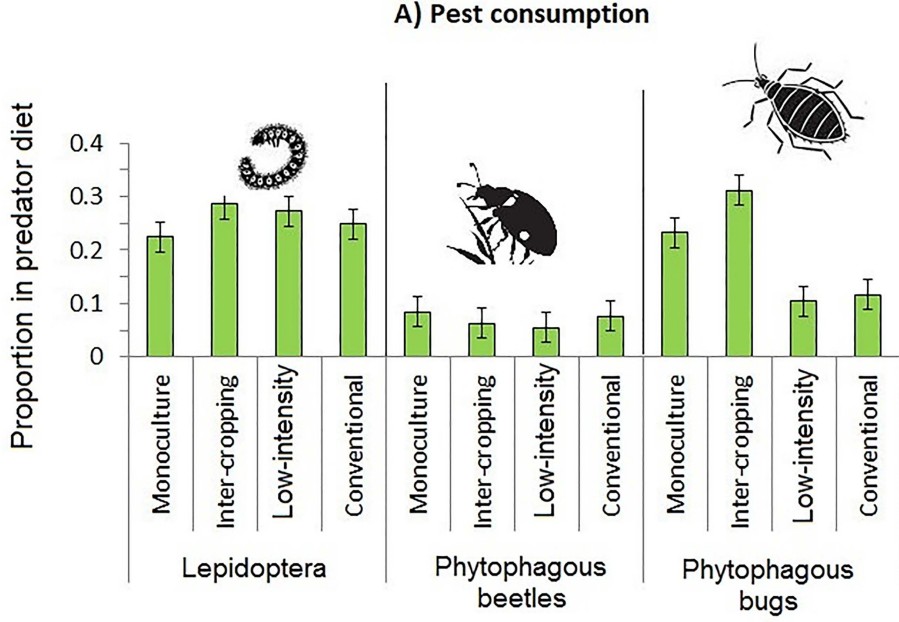

**A) Pest consumption**

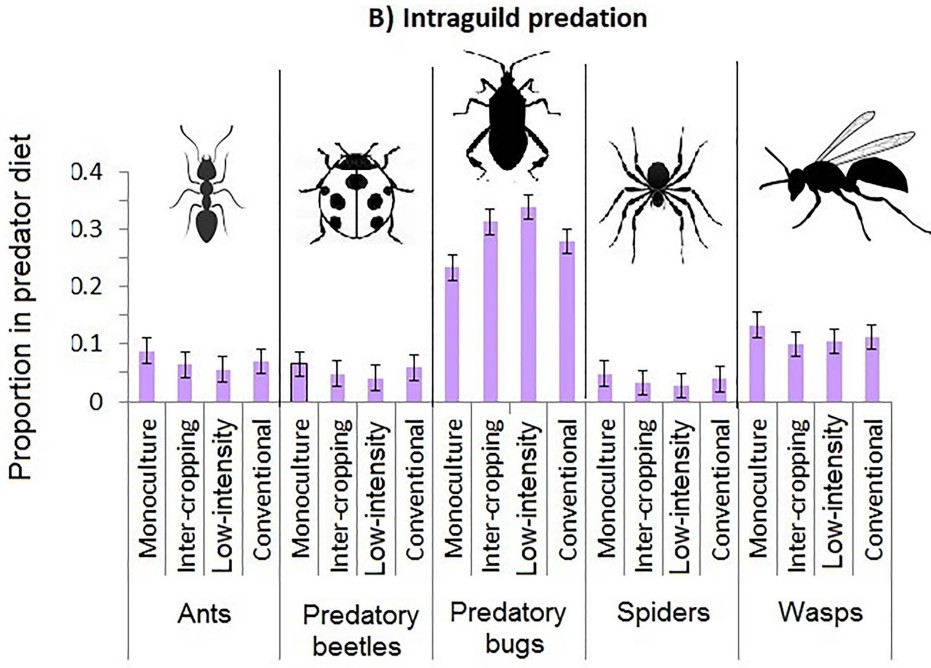

**B) Intraguild predation**

**Fig 5. Relationship between cropping method (monoculture vs intercropping) and farming system (low-intensity vs conventional) on overall consumption of the three pest and five natural enemy taxa by their predators.**

more vulnerable to predators by others within monoculture systems. By contrast, predatory bugs were consumed with proportionately higher intensively in low-intensity farms applying organic farming techniques whereas the other predators appeared more vulnerable to intraguild predation on conventional farms. Yet at the taxa level of pest consumption, there were higher rates of Lepidoptera consumption within low-intensity than conventional farming systems but no differences for the other two pests. Farming system further showed differential predictions for its influence on intraguild predation between predatory bugs and other predators, with more predatory bugs being consumed across low-intensity compared to conventional farming systems. While these observations are not easy to account for, it may be presumed that unlike cropping method, farming system's influence on predator feeding patterns, is manifested indirectly through other habitat factors not investigated in this study. For instance, low-intensity farming favoured consumption of the most common or abundant herbivorous (Lepidoptera) or intraguild prey (predatory bugs) whereas conventional farming systems pushed consumption patterns more strongly towards intraguild predation across the board.. This is consistent with findings by [79] that the most abundantly available pest prey, aphids tended to be targeted most by generalist predators in organically managed cereal fields. A closely related study by Galloway et al. [80] also showed that the benefits of low-intensity organic farming for promoting arthropod pest consumption appeared to be strongly linked to the nature of the surrounding landscape including the study farm's proximity to the crop-field in question, and its structural and qualitative attributes such as size and shape for maximizing predator spillovers. The authors observed, moreover, as did Gareau et al. [10] and Macfadyen et al. [81], that patterns of predator-pest interactions relating to the combined influence of crop-field's structural attributes and that of their surroundings, differed with pest-prey life histories as well as their distribution on temporal and spatial scales. Thus in the present study, the significantly higher Lepidoptera consumption in low-intensity farms may also arise from two possibilities. Firstly, such farming systems attract larger populations of both predatory and herbivorous arthropods [29]. Secondly, Lepidoptera are among the most abundant, common and detectable herbivores across crop-fields and hence more likely to be encountered and consumed by most predatory taxa [82].

Further, the higher rate of predatory bug consumption my other predators within low-intensity compared to conventional farming systems generally relates to the attractiveness of less intensively managed farms more abundant and diverse populations of arthropods in general and predatory taxa in particular [58,83,84]. Additionally, many predatory bug groups such as *Dicyphus* sp are omnivorous [83], and this may increase their own potential as intraguild prey. These trends imply, therefore, that low-intensity systems may serve to somewhat undermine the net functional potential of predatory bugs as pest regulators, particularly if their populations were to fall below a certain threshold due to such intraguild predation pressure.

### Implications for maize arthropod pest biological control

Predatory beetles' emergence as the most valuable pest consumer is consistent with many previous findings from which this taxon, especially ladybeetles (Coccinellidae) and carabid beetles (Carabidae) is widely recognized as important biological control agents [40,81] rather than as intraguild predators. As pointed out earlier on, the relatively substantial proportion of wasp carbon in predatory beetle diets may arise mainly from the latter's consumption of pest prey parasitized by these hymenopterans, and therefore this is unlikely to undermine wasps' overall roe in pest control. Similarly, ants' position as significant prey for spiders is unlikely to diminish these hymenopterans' overall contribution to pest consumption, given their resilience based on prolific reproduction, wide distribution, field-wide dispersal, and gregarious tendencies [20,30,47]. On the other hand, predatory bugs' apparent universal vulnerability as intraguild predation is likely to have significant negative impacts on their overall potential value as pest regulation agents, especially since their own trophic focus was oriented more towards intraguild predation than pest consumption.

In terms of habitat management strategy, only Lepidoptera pest removal was predicted to be potentially realistic through both low-intensity farming and maize intercropping, reaffirming these habitat management measures' significance in supporting natural enemies role in top-down suppression of this particular pest [26,27,80,85]. In particular, Soujanya et

al. [86] and Muneret et al. [29] showed clear linkage roles of intercropping and low intensity farming incorporating organic management, in promoting overall arthropod pest biocontrol. Conversely, with the exception of predatory bugs, the structurally simpler monoculture system appeared to be potentially more supportive of intraguild predation, which may be attributable to its association with low diversity of predatory arthropods and sometimes high dominance of a few dominant herbivore species [25,33,77]. Consequently, many specialist predators may be less attracted while intraguild predation pressure may increase thus undermining overall biological pest control [2,14,17]].

## Conclusion

This study has shown evidence of various levels and extents of intraguild predation amongst five of the most important arthropod pest natural enemy taxa in maize-fields. In particular, it has demonstrated that this form of trophic association is fundamentally exhibited by each of the five taxa examined here, with some of them especially predatory bugs and ants more frequently targeted than others. Secondly, farming systems and cropping methods are instrumental in driving these intraguild predatory as well as pest consumption patterns. Thus higher plot-level structural heterogeneity in maize farming, particularly intercropping or similar forms of crop diversification may be important in mediating reduced pressure of arthropod intraguild predation amongst pest natural enemies. Consequently through this, overall consumption rate of the most common pest taxa may be enhanced substantially. Intercropping maize with beans and general low-intensity farming may be particularly important for promoting removal of Lepidoptera and Hemiptera pests by wasps, predatory beetles and ants. Conversely, conventional farming options, characterized by more intensive management through agrochemical inputs and structurally simpler monoculture cropping, would more likely boost interactions amongst pest natural enemies thereby supporting higher rates of intraguild predation. Therefore maize-farm habitat management practices that promote diversity of crops at plot level and structural heterogeneity incorporating other structural elements at the landscape scale, can increase likelihood and efficiency of to-down pest suppression through biological control by mitigating negative impacts of intraguild predation

### The study's limitations and future research

This study did not account for predatory arthropods' food resources beyond the spatial limits of each study farms. For instance, the highly mobile wasps or the widely dispersing ants and carabid beetles [20,87] might obtain food from other farms or habitats outside the study farms, even if the pest prey themselves might be less mobile. However, potential errors arising from this are likely minimal as most farming landscapes in the study region are typically similar during the maize cropping season. Secondly, conclusions drawn here do not imply that these five were the only predatory arthropod taxa in these fields, or that these five may not themselves derive pest diet carbon through other predatory arthropods around the study site. Additionally, stable isotopic analyses results presented here may not have captured the full range of pest prey in predator diets beyond the three examined here, or temporal shifts in pest and intraguild prey choices during the period immediately preceding maize growing season (onset of cropping stage). Future studies within the study region should include not only analysis of interactive effects of cropping and farming system but also impact of mesopredation by other insectivorous taxa such as insectivorous birds, small mammals or reptiles, which might further impact predatory arthropods' roles in overall pest biocontrol. Quantification of proportions of predator diets contributed from scavenging and detritivorous and omnivorous feeding tendencies, would also facilitate further clarity on finer-scale arthropod trophic feeding linkages and their implications for overall top-down pest biocontrol across these maze-fields.

## Supporting information

**S1 Fig. MixSIAR model posterior density plots of 95% credible overlaps in patterns of consumption (diet proportions) of arthropod pest consumption and intraguild predation across the study farms.**
(TIF)

**S1 File. Checklist of herbivorous and predatory arthropod representative families from which samples were drawn for stable isotopic analyses, including their distribution across the farming systems and cropping methods.**
(PDF)

**S2 File. Topology of the selected study farms showing detailed attributes based on agronomic practice criteria: (farming system and cropping method).**
(PDF)

## Acknowledgments

We are very grateful to the farmers in Kakamega County, Kenya, who kindly allowed us into their farms to carry out this study, to the County, administrative Ward and village leaders for permitting out contact with the maize farmers and to the National Museums of Kenya for co-hosting the project and availing the project management, data analysis infrastructure support. Stellenbosch University and iThemba LABS Environmental Isotope Laboratory in South Africa are acknowledged for further assistance in project management and for undertaking stable isotope testing of field samples, respectively.

## Author contributions

**Conceptualization:** Nickson Erick Otieno.

**Data curation:** Nickson Erick Otieno, Jonathan Mukasi.

**Formal analysis:** Nickson Erick Otieno.

**Investigation:** Nickson Erick Otieno, Jonathan Mukasi.

**Methodology:** Nickson Erick Otieno.

**Project administration:** Nickson Erick Otieno.

**Software:** Nickson Erick Otieno.

**Supervision:** James Stephen Pryke.

**Validation:** James Stephen Pryke, Jonathan Mukasi.

**Visualization:** Jonathan Mukasi.

**Writing – original draft:** Nickson Erick Otieno.

**Writing – review & editing:** Nickson Erick Otieno, James Stephen Pryke.

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
