## [Decision Letter · Decision Letter 0]

18 Jun 2025

We look forward to receiving your revised manuscript.

Kind regards,

Muhammad Imran

Academic Editor

PLOS ONE

Journal Requirements:

3. We notice that your supplementary tables are included in the manuscript file. Please remove them and upload them with the file type 'Supporting Information'. Please ensure that each Supporting Information file has a legend listed in the manuscript after the references list.

**Additional Editor Comments:**

Thank you for submitting your manuscript entitled "Mitigating arthropod intraguild predation to enhance pest biocontrol better realized under maize-bean intercropping than through low-intensity farming." After careful evaluation of the peer review reports, it is clear that your study addresses a relevant topic in sustainable agriculture and pest management. However, both reviewers have identified several important issues that need to be addressed before the manuscript can be considered for publication. These include concerns related to the clarity and detail of the methodology, the statistical analysis, the interpretation of results, and the overall presentation and flow of the discussion. I encourage you to revise the manuscript thoroughly in line with the reviewers' comments, ensuring that all points are clearly addressed and justified. Please provide a detailed point-by-point response letter with your revised submission to facilitate re-evaluation. We look forward to receiving your revised manuscript

Reviewers' comments:

Reviewer's Responses to Questions

**Comments to the Author**

1. Is the manuscript technically sound, and do the data support the conclusions?

Reviewer #1: Yes

Reviewer #2: Yes

2. Has the statistical analysis been performed appropriately and rigorously?

Reviewer #1: Yes

Reviewer #2: No

3. Have the authors made all data underlying the findings in their manuscript fully available?

Reviewer #1: Yes

Reviewer #2: Yes

4. Is the manuscript presented in an intelligible fashion and written in standard English?

Reviewer #1: Yes

Reviewer #2: Yes

Reviewer #1: This study makes a strong contribution to understanding how farming practices influence intraguild predation (IGP) and arthropod-mediated pest control in maize systems. The application of stable isotope analysis to quantify trophic interactions is particularly innovative and adds significant ecological depth. Your core finding that maize-bean intercropping surpasses low-intensity farming alone in reducing IGP and enhancing pest suppression offers novel and highly relevant insights for sustainable agriculture, especially within smallholder contexts. To maximize the manuscript's clarity and impact for readers, several areas need refinement. The title and abstract require streamlining to enhance conciseness and explicitly highlight key quantitative outcomes. While the introduction provides solid background, it feels overly dense; sharpening the focus and concluding with a clearer, succinct statement of objectives and hypotheses would greatly improve its direction. Similarly, merging the results and discussion sections creates confusion; separating them would allow for clearer data presentation followed by distinct interpretation. Some ecological explanations, particularly those addressing underlying behavioral mechanisms, currently lean towards speculation and should be framed more cautiously as hypotheses for further testing. Finally, strengthening the conclusion to provide actionable guidance for farmers or policymakers on implementing intercropping would significantly enhance the study's practical relevance. Addressing these points will elevate an already valuable piece of research, ensuring its important findings on optimizing biocontrol through habitat management offer actionable insights for sustainable farming practices.

Reviewer #2: Reviewer Report

Manuscript Title:

Mitigating arthropod intraguild predation to enhance pest biocontrol better realized under maize-bean intercropping than through low-intensity farming

Manuscript Number:

PONE-D-25-26876

Recommendation:

Major Revision

General Assessment

This manuscript presents an important contribution to the field of agroecology and biological pest control by evaluating the role of habitat complexity (intercropping and low-intensity farming) in mitigating intraguild predation (IGP) among arthropod natural enemies using stable isotope analysis (SIA). The topic is timely and relevant, particularly for smallholder systems in the Global South.

The experimental approach using δ¹³C and δ¹⁵N isotopic signatures is well-justified, and the general structure of the manuscript is clear. However, significant improvements are required in hypothesis framing, description of methods, and interpretation of results to enhance scientific rigor and clarity.

Major Comments

1. Hypothesis Framing Needs Clarification

o The hypotheses presented are not clearly formulated or testable. The phrasing is long and ambiguous (lines 115–120).

o Recommendation: Clearly state the hypotheses at the end of the Introduction, using specific, testable language. For example:

“We hypothesized that (1) intercropping and low-intensity farming increase pest consumption and reduce intraguild predation, and (2) intraguild predation among predator taxa is not taxon-specific.”

2. Sampling Design Unclear and Unbalanced

o It is not clearly stated how many replicates exist per treatment combination (monoculture/intercrop × conventional/low-intensity), and the number appears unbalanced.

o Recommendation: Add a table summarizing the number of farms per treatment. Explain how unbalanced design, if present, was addressed analytically.

3. Statistical Comparisons Are Lacking

o While isotopic results are presented, there is no mention of credible interval overlap or formal tests of differences between treatments or taxa.

o Recommendation: Use posterior probability comparisons, credible interval overlap, or other Bayesian inference tools to validate statements of statistical difference.

4. Overinterpretation of SIA Results

o The manuscript often interprets stable isotope results as direct feeding relationships (e.g., scavenging vs. predation is not distinguished).

o Recommendation: Acknowledge this limitation in both Methods and Discussion. See Post (2002), Layman et al. (2012) for best practices.

Minor Comments

1. Grammar and Typographical Issues

o Several typographical errors throughout the paper (e.g., "concentualization", “low-intensityally”) detract from clarity.

o Recommendation: Thorough proofreading is required.

2. Figures and Tables

o Recommendation: Figure 1 quality needs to be improved to make it clear and readable.

3. Ethics and Limitations

o The “Limitations” section should better acknowledge potential external sources of error (e.g., unaccounted prey taxa, spatial spillover).

o Recommendation: Strengthen this section with a more balanced reflection on SIA scope and limitations.

4. Reference Formatting

o Several references contain inconsistencies or broken URLs. Few references like Blitzer et al 2012 have been included in bibliography but not cited.

o Recommendation: Carefully format references according to PLOS ONE guidelines.

**Do you want your identity to be public for this peer review?** For information about this choice, including consent withdrawal, please see our Privacy Policy

Reviewer #1: **Yes: ** Muhammad Imran

Reviewer #2: **Yes: ** Farrukh Baig

---

## [Author Response · Author response to Decision Letter 1]

11 Jul 2025

How Editorial and Review comments, concerns and suggestions have been addressed

Editor Comments:

Thank you for submitting your manuscript entitled "Mitigating arthropod intraguild predation to enhance pest biocontrol better realized under maize-bean intercropping than through low-intensity farming." After careful evaluation of the peer review reports, it is clear that your study addresses a relevant topic in sustainable agriculture and pest management. However, both reviewers have identified several important issues that need to be addressed before the manuscript can be considered for publication. These include concerns related to the clarity and detail of the methodology, the statistical analysis, the interpretation of results, and the overall presentation and flow of the discussion. I encourage you to revise the manuscript thoroughly in line with the reviewers' comments, ensuring that all points are clearly addressed and justified. Please provide a detailed point-by-point response letter with your revised submission to facilitate re-evaluation. We look forward to receiving your revised manuscript

AUTHORS’ RESPONSE: We hank he Editor for comments and the chance for us to revise and re-submit our article for re-evaluation. We have now revised te manuscript thoroughly by taking into account all the comments and concerns, and incorporating the suggested changes from the reviewers, presenting our responses in a point-by-point form according to the details appended further below and submitted along with our revised paper.

In summary, we have made the following major changes:

1. The title has now been modified to make it more concise and explicit about the study’s objectives and focus

2. The abstract has also been re-written with a clearer highlights of the key findings, especially the quantitative aspects of the results

3. The Results and Discussion sections have been separated so as to distinguish each of the findings (Results) from their respective interpretations further down, putting these within the context of other studies (Discussion)

4. We have re-written Discussion to make our recommendations less prescriptive

5. The hypotheses have now been more clearly/specifically and testably re-stated as follows:

6. We have now provided greater clarity on sampling protocol and balance in sample farm selectivity and design, as well as provided a new table to illustrate this

7. In analyses, we have now clarified the Bayesian statistical modeling tool MixSIAR, provided comparisons of observed patterns in isotopic test results for consumer-to-food source relations across the various attributes (predators and pests or intraguild prey) and also across levels of the various treatment factors (farming system and cropping method). In addition to and to more clearly and defensibly support these statistical results, we have now also included test results showing corresponding posterior probability plots, food proportion medians and 95% credible intervals (from lower limits of 2.5% to upper limits of 97.5%). These rae provided both in text as well as in tables and density plots, and now more clearly illustrate differences reported in Results and elaborated in Discussion.

8. We have now acknowledged that in addition to direct feeding interactions between predatory arthropod taxa and their pest and intraguild prey as revealed by isotopic test results, it is possible that some of the diet carbon may have been obtained through scavenging on carcasses of such prey, and also potentially from detritivorous feeding on decomposing tissue residues in the soil

9. We have now checked the articles and corrected for grammatical and typographic errors

10. We have now ensured that all references cited are listed in bibliography and vice versa

A more detailed account of our responses to review comments are presented below, with our responses highlighted in green

REVIEWERS' COMMENTS GENERAL:

Reviewer's Responses to Questions

Comments to the Author

1. Is the manuscript technically sound, and do the data support the conclusions?

Reviewer #1: Yes

Reviewer #2: Yes

2. Has the statistical analysis been performed appropriately and rigorously?

Reviewer #1: Yes

Reviewer #2: No

3. Have the authors made all data underlying the findings in their manuscript fully available?

Reviewer #1: Yes

Reviewer #2: Yes

4. Is the manuscript presented in an intelligible fashion and written in standard English?

Reviewer #1: Yes

Reviewer #2: Yes

5. Review Comments to the Author

Reviewer #1: Report

REVIEWER COMMENT/SUGGESTIONS: This study makes a strong contribution to understanding how farming practices influence intraguild predation (IGP) and arthropod-mediated pest control in maize systems. The application of stable isotope analysis to quantify trophic interactions is particularly innovative and adds significant ecological depth. Your core finding that maize-bean intercropping surpasses low-intensity farming alone in reducing IGP and enhancing pest suppression offers novel and highly relevant insights for sustainable agriculture, especially within smallholder contexts. To maximize the manuscript's clarity and impact for readers, several areas need refinement.

AUTHORS’ RESPONSE: We greatly appreciate this positive overall assessment by the Reviewer. The article has now been revised and edited according to the Refviewer’s recommendations, as detailed below

REVIEWER COMMENT/SUGGESTIONS: The title and abstract require streamlining to enhance conciseness and explicitly highlight key quantitative outcomes.

AUTHORS’ RESPONSE:

11. The title has now been modified to make it more concise and explicit about the study’s objectives and focus

12. The abstract has also been re-written with a clearer highlights of the key findings, especially the quantitative aspects of the results

REVIEWER COMMENT/SUGGESTIONS: While the introduction provides solid background, it feels overly dense; sharpening the focus and concluding with a clearer, succinct statement of objectives and hypotheses would greatly improve its direction.

AUTHORS’ RESPONSE: We have now re-written the Introduction in a way that is much under-densed and instead more closely focused on the subject matter, surveying relevant existing literature and placing our work where it fills a knowledge gap, before ending with a more succinct statement of our clearer objectives and what we hypothesized

REVIEWER COMMENT/SUGGESTIONS: Similarly, merging the results and discussion sections creates confusion; separating them would allow for clearer data presentation followed by distinct interpretation.

AUTHORS’ RESPONSE: The Results and Discussion sections have been separated so as to distinguish each of the findings (Results) from their respective interpretations further down, putting these within the context of other studies (Discussion)

REVIEWER COMMENT/SUGGESTIONS: Some ecological explanations, particularly those addressing underlying behavioral mechanisms, currently lean towards speculation and should be framed more cautiously as hypotheses for further testing.

AUTHORS’ RESPONSE: The reviewer did not name those explanations in specific, however, we have now re-written those parts the Discussion in where some of our arguments might have earlier tended towards speculation, and reworded them to be less finalistic, definitive, deterministic or prescriptive, and we have instead advanced standpoints that provide room for inviting further investigations on such postulations and interpreted positions

REVIEWER COMMENT/SUGGESTIONS: Finally, strengthening the conclusion to provide actionable guidance for farmers or policymakers on implementing intercropping would significantly enhance the study's practical relevance.

AUTHORS’ RESPONSE: The Conclusion subsection has now been restructured to give it a more prescriptive approach for farm managers and policymakers

REVIEWER COMMENT/SUGGESTIONS: Addressing these points will elevate an already valuable piece of research, ensuring its important findings on optimizing biocontrol through habitat management offer actionable insights for sustainable farming practices.

AUTHORS’ RESPONSE: We thank the reviewer for this positive verdict on the potential value our paper and the findings in it, and we hope that by have now addressed their concerns and incorporated the suggested changes to a sufficient extent.

Reviewer #2: Reviewer Report

Manuscript Title:

Mitigating arthropod intraguild predation to enhance pest biocontrol better realized under maize-bean intercropping than through low-intensity farming

Manuscript Number:

PONE-D-25-26876

REVIEWER COMMENT/SUGGESTIONS: Recommendation: Major Revision

AUTHORS’ RESPONSE: We greatly appreciate this positive overall assessment by the Reviewer. The article has now been revised and edited according to the Refviewer’s recommendations, as detailed below

REVIEWER COMMENT/SUGGESTIONS:

General Assessment

This manuscript presents an important contribution to the field of agroecology and biological pest control by evaluating the role of habitat complexity (intercropping and low-intensity farming) in mitigating intraguild predation (IGP) among arthropod natural enemies using stable isotope analysis (SIA). The topic is timely and relevant, particularly for smallholder systems in the Global South.

The experimental approach using δ¹³C and δ¹⁵N isotopic signatures is well-justified, and the general structure of the manuscript is clear. However, significant improvements are required in hypothesis framing, description of methods, and interpretation of results to enhance scientific rigor and clarity.

Major Comments

REVIEWER COMMENT/SUGGESTIONS: 1. Hypothesis Framing Needs Clarification

o The hypotheses presented are not clearly formulated or testable. The phrasing is long and ambiguous (lines 115–120).

o Recommendation: Clearly state the hypotheses at the end of the Introduction, using specific, testable language. For example:

“We hypothesized that (1) intercropping and low-intensity farming increase pest consumption and reduce intraguild predation, and (2) intraguild predation among predator taxa is not taxon-specific.”

AUTHORS’ RESPONSE: The hypotheses have now been more specifically and testably re-stated as follows: “We hypothesized that 1) Intercropping and low-intensity farming increase pest consumption and reduce intraguild predation pressure; 2) Prey selectivity amongst predator taxa is not taxon-specific”.

REVIEWER COMMENT/SUGGESTIONS: 2. Sampling Design Unclear and Unbalanced

o It is not clearly stated how many replicates exist per treatment combination (monoculture/intercrop × conventional/low-intensity), and the number appears unbalanced.

o Recommendation: Add a table summarizing the number of farms per treatment. Explain how unbalanced design, if present, was addressed analytically.

AUTHORS’ RESPONSE: We have stated more clearly that the study farms were a total of 18, and of these, 3 were monoculture organic farms, 5 were intercropped organic, 4 were monoculture conventional and 3 were intercropped conventional. Therefore, although through this combination, the 7 conventional versus 8 organic were counterbalanced by the 8 intercropped vs 7 monoculture. Our analyses were based on comparing organic to conventional (total 15) and intercropped to monocropped ((total 15), rather than the interactive pairwise interplays between these treatments, which is the only case in which balancing might become a statistical concern. In addition, any potential balance-related statistical biases were further minimized by the fact that sampling was conducted at several times (multiple crop stages and multiple cropping seasons), which were then ultimately averaged/pooled rather than analyzed separately. We have now also added a supplementary/supporting information table (S2_File) outlining the details of study farm biotopes

REVIEWER COMMENT/SUGGESTIONS: 3. Statistical Comparisons Are Lacking

o While isotopic results are presented, there is no mention of credible interval overlap or formal tests of differences between treatments or taxa.

o Recommendation: Use posterior probability comparisons, credible interval overlap, or other Bayesian inference tools to validate statements of statistical difference.

AUTHORS’ RESPONSE: To add to the credible intervals for results that we already presented from MixSIAR Bayesian modeling in Table 1 for most significant intraguild predation patterns (diet contribution levels), we have now also added two new tables (Table 2 and Table 3) for estimated patterns of overall contributions of each prey (pest or predator) to diets of their consumers, together with respective values for medians and 95% credible intervals (lower as well as upper limits). We have also added a supplementary figure (S1 Fig) to the supporting material, showing the MixSIAR model posterior density plots, complete with the various patterns of 95% credible interval overlaps in food source contributions to the diets of their respective consumers Table 3, in particular, indcates the effects of the treatments (farming system and cropping method. Due to the large volume of test results usually generated from a typical MixSIAR Bayesian modeling (tens of tables and figures at a time), we are not able to present all statistical data in the results. Therefore we hope the Reviewer would concur with us that as a form of Bayesian modeling procedure, MixSIAR is a formal statistical analysis in itself for predicting probability distributions (in this case diverse food-source options contributing to diets of consumers) different from/alternative to the classical goodness-of-fit-to-mean approaches testing for p-values within specified confidence limits. Therefore for MixSIAR, posterior density overlap plots of the estimated median probabilities, accompanied by 95% credible intervals, which we have now added, should be sufficiently defensible.

REVIEWER COMMENT/SUGGESTIONS: 4. Overinterpretation of SIA Results

o The manuscript often interprets stable isotope results as direct feeding relationships (e.g., scavenging vs. predation is not distinguished).

o Recommendation: Acknowledge this limitation in both Methods and Discussion. See Post (2002), Layman et al. (2012) for best practices.

AUTHORS’ RESPONSE: We have now duly acknowledged in both Methods, and (right at the state of Discussion) sections, that scavenging might have added to the overall diet carbon intake by intraguild predators, bu

---

## [Decision Letter · Decision Letter 1]

22 Jul 2025

Maize-bean intercropping mediates reduction in arthropod intraguild predation better than low-intensity farming – stable isotope evidence

PONE-D-25-26876R1

We’re pleased to inform you that your manuscript has been judged scientifically suitable for publication and will be formally accepted for publication once it meets all outstanding technical requirements.

Kind regards,

Muhammad Imran

Academic Editor

PLOS ONE

Additional Editor Comments (optional):

I am pleased to inform you that your manuscript, entitled “Maize-bean intercropping mediates reduction in arthropod intraguild predation better than low-intensity farming: stable isotope evidence,” has been accepted for publication. Following the second round of peer review, both reviewers have recommended acceptance, indicating that the revisions you made adequately addressed their previous concerns. The decision to accept your manuscript is based on the positive evaluations provided by the reviewers. We appreciate your efforts in revising the manuscript and incorporating the reviewer’s suggestions. Please proceed with any final formatting or submission requirements as outlined in our journal’s guidelines. Congratulations on your successful submission, and publishing your work

Reviewers' comments:

Reviewer's Responses to Questions

**Comments to the Author**

Reviewer #1: All comments have been addressed

Reviewer #2: All comments have been addressed

2. Is the manuscript technically sound, and do the data support the conclusions?

Reviewer #1: Yes

Reviewer #2: Yes

3. Has the statistical analysis been performed appropriately and rigorously?

Reviewer #1: Yes

Reviewer #2: Yes

4. Have the authors made all data underlying the findings in their manuscript fully available?

Reviewer #1: Yes

Reviewer #2: Yes

5. Is the manuscript presented in an intelligible fashion and written in standard English?

Reviewer #1: Yes

Reviewer #2: Yes

Reviewer #1: After reviewing the revised manuscript titled "Maize-bean intercropping mediates reduction in arthropod intraguild predation better than low-intensity farming stable isotope evidence," I am pleased to note that the authors have thoroughly addressed all the suggestions and shortcomings identified during the initial review. The revisions incorporated are comprehensive and effectively resolve the concerns raised, enhancing the clarity, rigor, and overall quality of the manuscript. I believe the manuscript is now well-suited for publication.

Reviewer #2: (No Response)

**Do you want your identity to be public for this peer review?** For information about this choice, including consent withdrawal, please see our Privacy Policy

Reviewer #1: **Yes: ** Muhammad Imran

Reviewer #2: **Yes: ** Farrukh Baig

---

## [Editor Report · Acceptance letter]

PONE-D-25-26876R1

PLOS ONE

Dear Dr. Otieno,

I'm pleased to inform you that your manuscript has been deemed suitable for publication in PLOS ONE. Congratulations! Your manuscript is now being handed over to our production team.

Kind regards,

on behalf of

Dr. Muhammad Imran

Academic Editor

PLOS ONE